# Site-specific cleavage of bacterial MucD by secreted proteases mediates antibacterial resistance in *Arabidopsis*

Yiming Wang [1], Ruben Garrido-Oter[1,2], Jingni Wu [1,4], Thomas M. Winkelmüller[1], Matthew Agler [1,5], Thomas Colby[3,6], Tatsuya Nobori[1], Eric Kemen[1,7] & Kenichi Tsuda [1]

Plant innate immunity restricts growth of bacterial pathogens that threaten global food security. However, the mechanisms by which plant immunity suppresses bacterial growth remain enigmatic. Here we show that *Arabidopsis thaliana* secreted aspartic protease 1 and 2 (SAP1 and SAP2) cleave the evolutionarily conserved bacterial protein MucD to redundantly inhibit the growth of the bacterial pathogen *Pseudomonas syringae*. Antibacterial activity of SAP1 requires its protease activity in planta and in vitro. Plants overexpressing *SAP1* exhibit enhanced MucD cleavage and resistance but incur no penalties in growth and reproduction, while *sap1 sap2* double mutant plants exhibit compromised MucD cleavage and resistance against *P. syringae*. *P. syringae* lacking *mucD* shows compromised growth in planta and in vitro. Notably, growth of *ΔmucD* complemented with the non-cleavable MucD$^{F106Y}$ is not affected by SAP activity in planta and in vitro. Our findings identify the genetic factors and biochemical process underlying an antibacterial mechanism in plants.

[1] Department of Plant Microbe Interactions, Max Planck Institute for Plant Breeding Research, Carl-von-Linne Weg 10, 50829 Cologne, Germany. [2] Cluster of Excellence on Plant Sciences, Heinrich Heine University Düsseldorf, 40225 Düsseldorf, Germany. [3] Plant Proteomics Group, Max Planck Institute for Plant Breeding Research, Carl-von-Linne Weg 10, 50829 Cologne, Germany. [4] Present address: Institute of Plant Physiology and Ecology, Chinese Academy of Sciences, 200032 Shanghai, China. [5] Present address: Plant Microbiosis Lab, Institute of Microbiology, Friedrich-Schiller University Jena, Neugasse 23, 07743 Jena, Germany. [6] Present address: Max Planck Institute for Biology of Ageing, Joseph-Stelzmann-Strasse 9B, 50931 Cologne, Germany. [7] Present address: Center for Plant Molecular Biology, Interfaculty Institute of Microbiology and Infection Medicine Tübingen, University of Tübingen, Auf der Morgenstelle 32, 72076 Tübingen, Germany. Correspondence and requests for materials should be addressed to K.T. (email: tsuda@mpipz.mpg.de)

During the course of co-evolution with microbial pathogens, plants and animals have evolved highly sophisticated innate immune systems to defend themselves against pathogen infection[1,2]. In both systems, specific host receptors recognize microbial molecules, thereby activating cellular signaling pathways that eventually contribute to the suppression of pathogen growth[3–5]. The immune system needs to be tightly controlled because over activation causes autoimmune diseases in humans and often involves growth retardation in plants. This so-called immunity-growth tradeoff in plants poses a dilemma in agriculture[6–8].

Activation of host immune signaling pathways leads to the production of antimicrobial molecules and cellular changes in the host that directly alters microbial metabolisms, resulting either in pathogen growth suppression or demise[2,9]. In animals, circulating immune cells exert antibacterial activity through multiple extensively studied mechanisms[10,11], such as direct bacterial killing, activation of antimicrobial peptides, and attenuation of bacterial virulence by secreted proteases[12,13]. On the other hand, evidence for how plants suppress bacterial growth is rather limited. Plants produce antimicrobial peptides or secondary metabolites, which have antibacterial properties in vitro, but their physiological relevance and modes of action in plants remain obscure[14].

Many plant proteases are predicted to be secreted into the extracellular space (apoplast), which is an important niche for leaf bacterial pathogens[15–19]. Some secreted proteases play roles in plant immunity. For instance, the extracellular subtilase SBT3.3 positively contributes to resistance against bacterial and fungal pathogens in Arabidopsis thaliana as the absence or overexpression of SBT3.3 leads to susceptibility or resistance, respectively[17]. Similarly, a secreted aspartic protease CDR1 is an important player during immunity against bacterial pathogens in A. thaliana as well as in rice[20,21]. Pip1 is a secreted papain-like protease that contributes to resistance in tomato against pathogens across multiple kingdoms[19]. Plant pathogens produce protease inhibitors to counteract the host proteases, supporting the idea that plants and pathogens engage in protease warfare on the battleground in the apoplast[22]. Although the studies above have provided strong evidence that secreted proteases are important components of plant immunity, both SBT3.3 and CDR1 carry out their roles by activating plant immune signaling pathways[17,21], and their target for immunity remains unknown[22].

In the present study, we provide compelling biochemical and genetic evidence that A. thaliana secretes the secreted aspartic protease 1 (SAP1) and SAP2 to cleave the bacterial protein MucD, thereby suppressing Pseudomonas syringae growth in the leaf apoplast. Both SAP and mucD are evolutionarily conserved in the plant and bacterial kingdoms, respectively. Our work, therefore, sheds light on the previously poorly understood mechanisms by which plants protect themselves against bacterial pathogens.

## Results

**SAP1 and SAP2 suppress *P. syringae* growth in planta.** Foliar bacterial pathogens colonize the extracellular space, and thus to gain insights into how plant immunity suppresses bacterial growth in the leaf apoplast, we tested the ability of immune-activated apoplastic fluid (from leaves treated with the flg22 peptide from bacterial flagellin) to suppress bacterial growth in vitro. This apoplastic fluid suppressed growth of *P. syringae* pv. tomato DC3000 (*Pto*) compared to that from water-treated leaves, and this effect was heat sensitive (Fig. 1a), implying that a protein is responsible. We reasoned that secreted proteases are plausible candidates as they would be able to directly target bacterial protein in the apoplast. The *A. thaliana* genome contains over

700 genes encoding putative proteases[23]. We focused on aspartic proteases as they generally have optimum activity at the acidic pH of the plant apoplast[24–26]. We found 77 *A. thaliana* aspartic proteases in MEROPS[27] (Supplementary Fig. 1A). Sixty-one possess an N-terminal secretion signal peptide[28], and 51 are predicted to have extracellular localization in TAIR10[29]. Of these, two tandemly arrayed genes (*At1g03230* and *At1g03220*), whose expression was not distinguishable by microarray-based system due to their high sequence similarity in Genevestigator, show consistent induction by both flg22 and *Pto*[30] (Supplementary Fig. 1A). Since these genes have not been previously described, we termed them *SAP1* and *SAP2*, respectively.

We determined the individual expression levels of *SAP1* and *SAP2* by quantitative reverse transcription PCR (RT-qPCR). Both *SAP1* and *SAP2* expression was induced upon flg22 treatment and *Pto* infection (Fig. 1b). Immunoblotting of the SAP fusion proteins showed slightly increased apoplastic accumulation upon *Pto* infection (Fig. 1c). To test if the apoplastic localization was signal peptide dependent, full-length *SAP1-RFP* or *SAP2-RFP* or signal peptide-depleted *SAP1 (ΔSP)-RFP* or *SAP2 (ΔSP)-RFP*, driven by a constitutive ubiquitin promoter, were transiently expressed in *A. thaliana* transgenic plants expressing a plasma-membrane-localized GFP (green fluorescent protein) (WAVE131)[31]. RFP (red fluorescent protein) signals were detected between GFP signals in a signal-peptide-dependent manner (Fig. 1d), suggesting that SAP1 and SAP2 are secreted into the apoplast via the canonical protein secretion pathway[32,33].

Two independent T-DNA insertion mutants for *SAP1*, SALK_062079 (*sap1-1*) and SAIL_646_E08 (*sap1-2*), were obtained, and disruption of *SAP1* was confirmed in both lines (Supplementary Fig. 1B, C). We also generated *SAP2* RNA interference (RNAi) and CRISPR-Cas9-knockout lines in wild-type Col as well as in *sap1-1* background since no *sap2* T-DNA insertion lines were available (Supplementary Fig. 1D, E). Only *sap1 sap2* CRISPR-Cas9 knockout (*sap1 sap2*) and *sap1 SAP2*-RNAi plants exhibited increased susceptibility to *Pto*, while single mutant and *SAP2*-RNAi plants did not, implicating that *SAP1* and *SAP2* redundantly contribute to resistance against *Pto* (Fig. 2a and Supplementary Fig. 1F). We also generated transgenic *A. thaliana* plants, which constitutively express either the full-length *SAP1* or *SAP2* or *SAP1ΔSP* or *SAP2ΔSP*, all of which were generated as fusion proteins with RFP at the C terminus. We observed decreased bacterial growth in transgenic plants expressing the full-length *SAP1-RFP* or *SAP2-RFP* as compared with wild-type Col plants but not in plants expressing *SAP1ΔSP* or *SAP2ΔSP*, which localized to the cytosol (Figs. 1d, 2b). Immunoblotting confirmed that protein expression levels were comparable in all transgenic lines (Fig. 2b). Thus, the apoplastic localization of SAP1 and SAP2 is essential for bacterial growth suppression. Enhanced disease resistance is often associated with constitutive immune activation and a consequent growth penalty[6–8]. Interestingly, none of transgenic plants showed enhanced expression of the immune marker *PR1*, except for *pUB::SAP2* line 2 with a slight increase, growth retardation, or reduced reproduction, but some of them showed enhanced growth and reproduction (Fig. 2c–f). These data imply that SAP1 and SAP2 influence bacterial proliferation via direct interaction with bacteria in the apoplast, and that their overexpression poses minimal plant fitness costs.

**SAP1 and SAP2 suppress *P. syringae* growth in vitro.** To test if SAP1 and SAP2 suppress bacterial growth by direct interaction, we produced in *Escherichia coli* and purified recombinant SAP1ΔSP and SAP2ΔSP fused to GST at the C terminus. As compared with the GFP control, both SAP1 and SAP2 showed

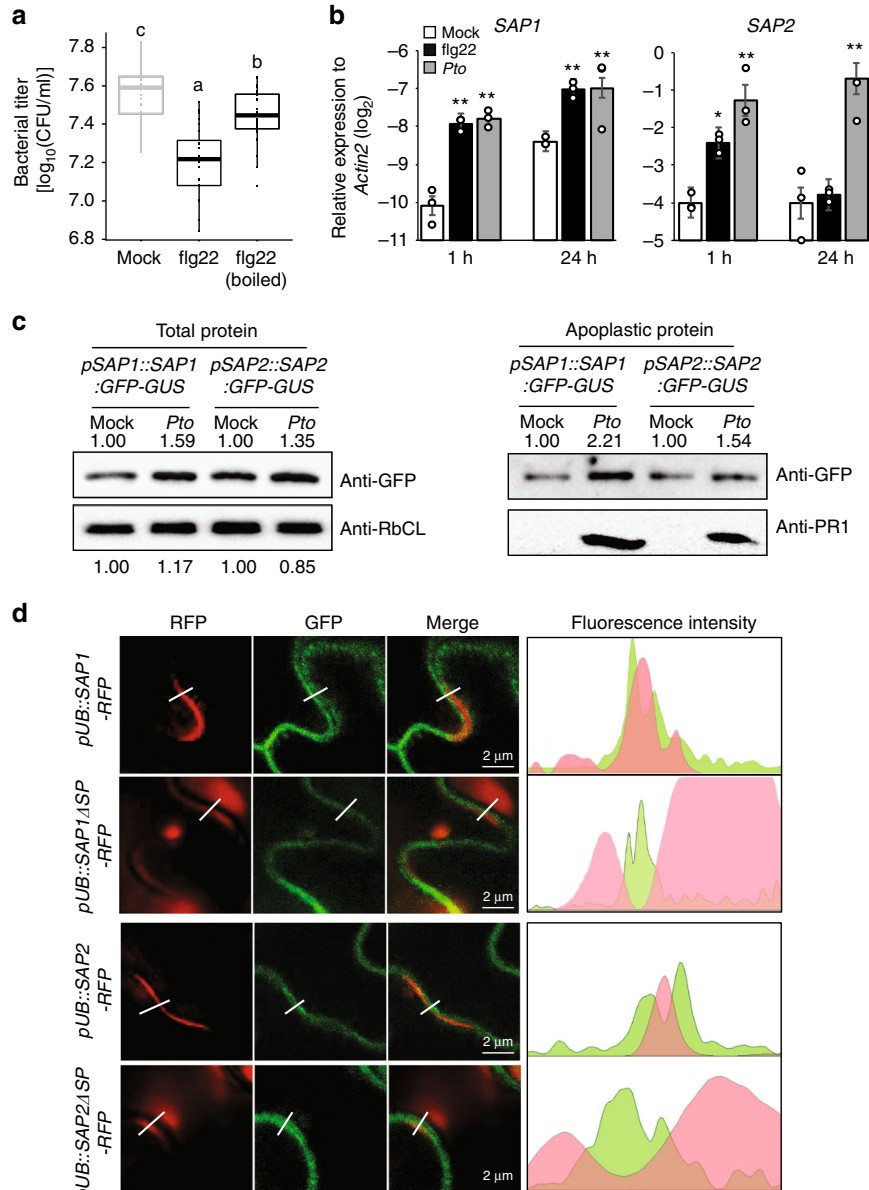

**Fig. 1** Secreted aspartic protease 1 (SAP1) and SAP2 are induced and secreted during *P. syringae* infection. **a** In vitro growth of *Pto* (OD$_{600}$ = 0.05) amended with apoplastic fluid from leaves of -week-old Col plants at 24 h post infiltration (hpi) with 1 μM flg22 or mock or boiled apoplastic fluid (boiled) was measured at 9 h after culturing. Bars represent means and s.e.m. of three independent experiments each with eight replicates. Statistically significant differences are indicated by different letters (adjusted $P < 0.05$). **b** Expression of *SAP1* and *SAP2* in leaves of 4-week-old Col plants at 1 or 24 hpi with mock, 1 μM flg22, or *Pto* (OD$_{600}$ = 0.001). Bars represent means and s.e.m. of three biological replicates. Asterisks indicate significant differences (Student's two-tailed *t* test; *$P < 0.05$, **$P < 0.01$). **c** Accumulation of SAP1-GFP (green fluorescent protein) and SAP2-GFP at 1 day post infiltration (dpi) with *Pto* (OD$_{600}$ = 0.001) or mock determined by immunoblotting using an anti-GFP antibody. Rubisco large subunit (RbCL) and PR1 serve as controls for total and apoplastic proteins, respectively. **d** SAP1-RFP (red fluorescent protein) and SAP2-RFP (red color) with or without the signal peptide (ΔSP) were expressed from the 35S promoter by *Agrobacterium*-mediated transient transformation in transgenic *A. thaliana* plants expressing plasma-membrane-localized WAVE131-YFP (yellow fluorescent protein) (green color). YFP and RFP fluorescence signals were detected at 2 dpi. The intensity of YFP and RFP fluorescence signals was quantified along the dotted lines using ImageJ software (left to right). Four independent experiments were performed with similar results

protease activity, which was blocked by an aspartic protease inhibitor, pepstatin A (Fig. 3a). These active SAP1 and SAP2 proteins, but not the heat-inactivated or GFP controls, suppressed in vitro *Pto* growth (Fig. 3b). In MEROPS, SAP1 and SAP2 are classified into Clan AA Family A1, which includes an *A. thaliana* aspartic protease involved in immune activation, CDR1[21,34]. Aspartic proteases generally require two conserved Asp residues that intramolecularly form the catalytically active site, although some are known to form a homodimer that intermolecularly

generates an active site consisting of two Asp residues[35]. SAP1 and SAP2 are described as "non-peptidase homologs," as they lack one of the Asp residues in the active site. Sequence alignment revealed that one of the catalytically essential Asp residues in CDR1 is replaced with Ser in SAP1 and SAP2 (Supplementary Fig. 2A, B). To understand the evolutionary conservation of SAP1 and SAP2 protein sequences, we searched for SAP1 and SAP2 homologs in the Brassicaceae family, to which *A. thaliana* belongs, as well as in the more distant plant species, tomato and

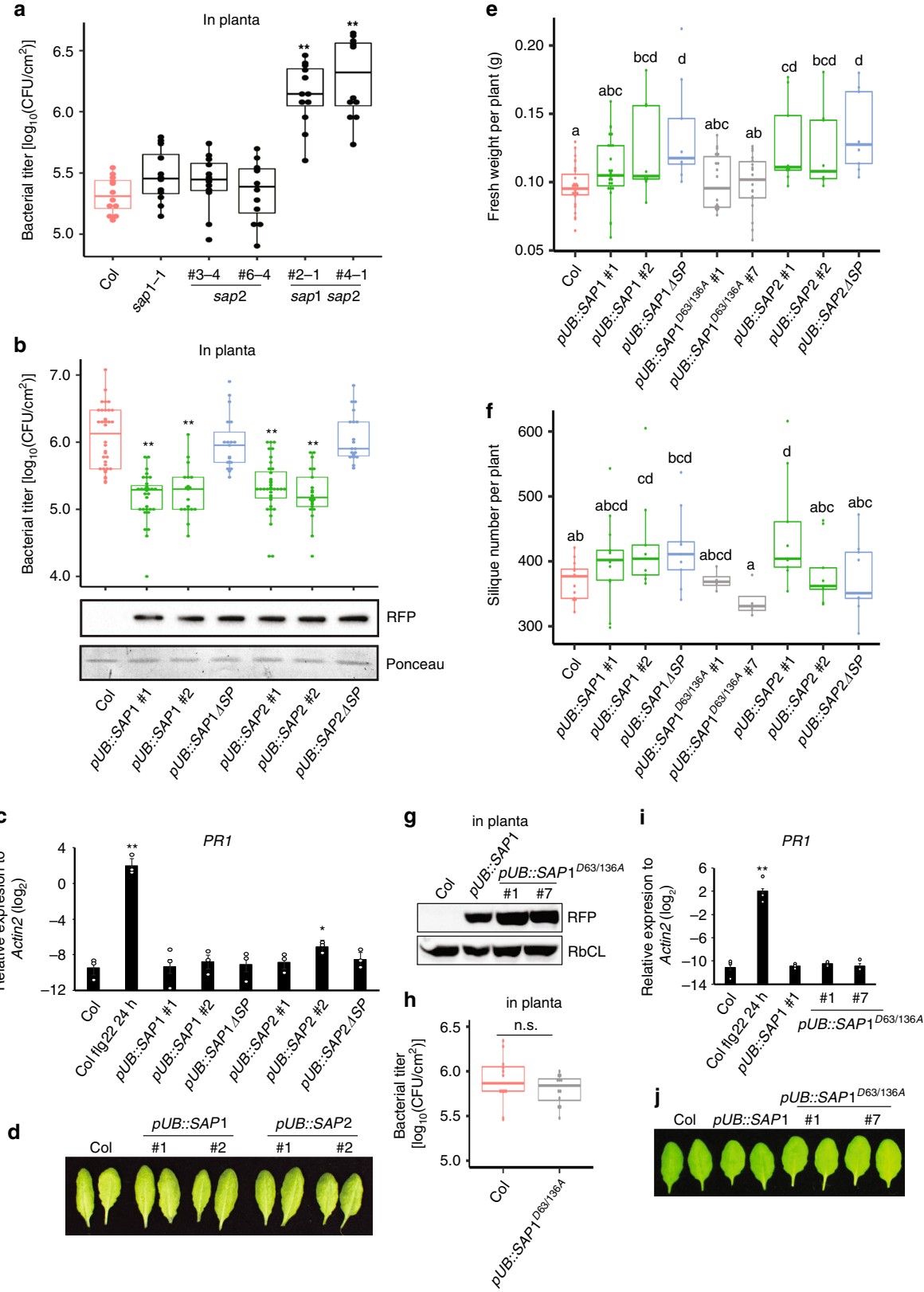

rice. All analyzed species have one to three *SAP* homologs (Supplementary Fig. 3A, B), and expression of most of these genes was induced by flg22 (rice was not tested) (Supplementary Fig. 3C), suggesting that the importance of SAPs in antibacterial defense is evolutionarily conserved. Seven Asp residues including one in the active site and the substituted Ser are conserved in all

*SAP* homologs (Supplementary Fig. 2A and Fig. 3a), pointing to the importance of these residues for SAP function. We produced recombinant SAP1 variants in which these eight amino acid residues were substituted with Ala and tested them for protease and in vitro bacterial suppression activity (Supplementary Fig. 2C-E). Similar to the GFP control, D63A and D63/136A

**Fig. 2** Secreted aspartic protease 1 (*SAP1*) and *SAP2* redundantly contribute to resistance against *P. syringae*. **a** Leaves of 4-week-old Col, *sap1*, *sap2*, and *sap1 sap2* double mutant plants were infiltrated with *Pto* ($OD_{600} = 0.0005$), and bacterial titer was determined at 3 days post infection (dpi). Bars represent means and s.e.m. of four independent experiments with three biological replicates. Asterisks indicate significant differences from Col (Student's two-tailed *t* test; *$P < 0.05$, **$P < 0.01$). **b** Leaves of 4-week-old *pUB::SAP1-RFP* (red fluorescent protein) and *SAP2-RFP* with or without signal peptide (*ΔSP*) were infiltrated with *Pto* ($OD_{600} = 0.001$), and bacterial titer was measured at 2 dpi. Bars represent means and s.e.m. of three independent experiments with at least three biological replicates. Asterisks indicate significant differences from Col (Student's two-tailed *t* test; *$P < 0.05$, **$P < 0.01$). Protein expression determined by immunoblotting using an anti-RFP antibody. Ponceau S staining serves as protein loading control. **c, i** Relative expression of *PR1* in leaves of 4-week-old Col, *pUB::SAP1-RFP*, *pUB::SAP1ΔSP-RFP*, *pUB::SAP2-RFP*, *pUB::SAP2ΔSP-RFP*, and *pUB::SAP1^{D63/136A}* plants were determined by quantitative reverse transcription PCR (RT-qPCR). Bars represent means and s.e.m of three biological replicates. The vertical axis shows the $\log_2$ expression levels relative to *Actin2*. Col plants 24 h post infiltration (hpi) with flg22 serves as a positive control for activated *PR1* expression. Asterisks indicate significant differences from Col (Student's two-tailed *t* test; *$P < 0.05$, **$P < 0.01$). **d, j** Phenotype of 4-week-old leaves of Col, *pUB::SAP1-RFP* and *pUB::SAP2-RFP*, and *pUB::SAP1^{D63/136A}* plants. Shoot fresh weight (4-week-old; **e**) and silique numbers (**f**) of Col, *pUB::SAP1-RFP*, *pUB::SAP1ΔSP-RFP*, *pUB::SAP1^{D63/136A}-RFP*, *pUB::SAP2-RFP*, and *pUB::SAP2ΔSP-RFP* plants. Bars represent means and s.e.m. of three independent experiments each with four biological replicates (**e**) or means and s.e.m. of three independent experiments (**f**). Statistically significant differences are indicated by different letters (adjusted $P < 0.01$). **g** Protein expression of SAP1 and SAP1^{D63/136A} in Col, *pUB::SAP1-RFP* and *pUB::SAP1^{D63/136A}-RFP* plants by immunoblotting using an anti-RFP antibody. Rubisco large subunit (RbCL) serves as a loading control. Three independent experiments were performed with similar results. **h** Leaves of 4-week-old Col and *pUB::SAP1^{D63/136A}* plants were infiltrated with *Pto* ($OD_{600} = 0.001$), and bacterial titer was measured at 2 dpi. Bars represent means and s.e.m, of four independent experiments with four biological replicates. n.s., not significant

SAP1 variants showed no protease activity, although protein levels were similar to the other variants (Fig. 3c and Supplementary Fig. 2C, E). Notably, these inactive variants did not suppress in vitro *Pto* growth (Fig. 3d and Supplementary Fig. 2D), indicating that SAP1 protease activity is essential for its function in in vitro *Pto* growth suppression. Overexpression of the non-active variant SAP1^{D63/136A} in plants, which did not influence *PR1* expression, plant growth, and reproduction (Fig. 2e–j), had no effect on plant resistance against *Pto* (Fig. 2h). Taken together, these results demonstrate that SAP1, annotated as a pseudo-peptidase, functions as an active aspartic protease that suppresses *Pto* growth in vitro and in planta via its protease activity.

**SAP1 cleaves *Pto* MucD**. To elucidate how SAP1 suppresses *Pto* growth, in vitro bacterial culture was incubated with SAP1-GST or heat-inactivated SAP1-GST, and proteins from bacterial cells and the medium were separated by gel electrophoresis. Although there were no apparent differences in the band pattern of proteins from bacterial cells, we observed that the intensity of one band of a molecular weight of ~50 kDa in the medium was reduced specifically by incubation with the active SAP1-GST (Fig. 3e). The corresponding protein band was subjected to liquid chromatography with tandem mass spectrometry (LC-MS/MS) analysis to identify SAP1-target *Pto* proteins (SAPTs). Mapping of detected peptides to the *Pto* proteome resulted in 21 SAPT candidates with a reduced peptide levels in the active SAP1-GST sample (Supplementary Table 1). The top SAPT candidate containing a putative aspartic protease digestion site(s) was MucD, which is an HtrA-like protease involved in the regulation of alginate bio-synthesis and in the responses to heat and oxidative stress[36,37]. A MucD homolog in the human opportunistic bacterial pathogen *Pseudomonas aeruginosa* is known to function as a serine pro-tease and is important for virulence in animals as well as in plants[36,38]. An in vitro cleavage assay showed that SAP1 and SAP2 could cleave MucD, but that the protease-dead variant SAP1^{D63/136A} and the GFP control could not (Fig. 3f).

To investigate whether MucD cleavage occurs in planta, *Pto* expressing MucD-HA was infiltrated into leaves of Col plants, and MucD cleavage was monitored. The cleaved MucD product was detected in Col plants (Fig. 3g). The detected bands were specific to MucD-HA as the bands were not detected using protein extracted from plants infected with wild-type *Pto* (Supplementary Fig. 4A). MucD cleavage was enhanced in *pUB::SAP1-RFP* plants as compared to Col and *pUB::SAP1^{D63/136A}-RFP* plants (Fig. 3g and Supplementary Fig. 4B). MucD cleavage was slightly reduced

in single *sap1* and *sap2* mutant and *sap1 SAP2-RNAi* plants (Supplementary Fig. 4C, D), and almost undetectable in *sap1 sap2* CRISPR double mutant plants (Fig. 3h). These results indicate that SAP1 and SAP2 are required to cleave *Pto* MucD during infection. The minor MucD cleavage detected in *sap1 sap2* mutant plants suggests that SAP1 and SAP2 are the major enzymes that cleave *Pto* MucD, while other enzymes might also be involved in MucD cleavage.

**mucD is required for in planta *Pto* growth**. To determine whether *mucD* is important for *Pto* growth, *mucD* deletion mutants (*ΔmucD*) and complemented lines (MucD-HA) of *Pto* were generated. *Pto ΔmucD* showed a mucoid phenotype prob-ably due to overproduction of alginate[39], and the complementa-tion line exhibited a wild-type-like phenotype (Supplementary Fig. 5A). We found that *Pto* MucD localized to the membrane, but was also secreted outside the cell (Supplementary Fig. 5B) as in *P. aeruginosa*[38,39]. Compared to wild-type *Pto*, in vitro growth of *Pto ΔmucD* was slower and in planta growth was severely compromised, phenotypes that were both rescued by MucD-HA complementation (Fig. 4d–g and Supplementary Fig. 5C, D). These results indicate that *mucD* is required for optimal *Pto* growth in vitro and in planta.

**SAP1 and SAP2 suppress *Pto* growth via MucD cleavage**. Based on in silico analysis, the MucD protein sequence was predicted to harbor two putative aspartic protease cleavage sites[40] (Fig. 4a). We generated MucD^{F106Y} and MucD^{S394A} with a mutation at each of the putative cleavage sites by aspartic proteases and produced His-tag-fused recombinant proteins at the C terminus. In the in vitro cleavage assay, we observed that SAP1-GST cleaves MucD^{S394A} but not MucD^{F106Y} (Fig. 4b). This is congruent with the size of the cleaved MucD product in vitro and in planta (Figs. 3f–h, 4b, c). To test whether this holds true in planta, we generated *ΔmucD* lines complemented with MucD^{F106Y}. We inoculated Col, *pUB::SAP1-RFP*, and *pUB::SAP1^{D63/136A}-RFP* plants with wild-type *Pto* and *Pto ΔmucD* MucD^{F106Y} and detected cleavage of wild-type MucD but not MucD^{F106Y} (Fig. 4c). These results indicate that F106 is the amino acid residue in MucD that is critical for cleavage by SAP1 in vitro and in planta.

MucD^{F106Y}-HA but not MucD^{S394A}-HA complemented the mucoid phenotype and in vitro growth of *Pto ΔmucD*, indicating that MucD^{F106Y} is functional (Fig. 4d and Supplementary Fig. 5A,

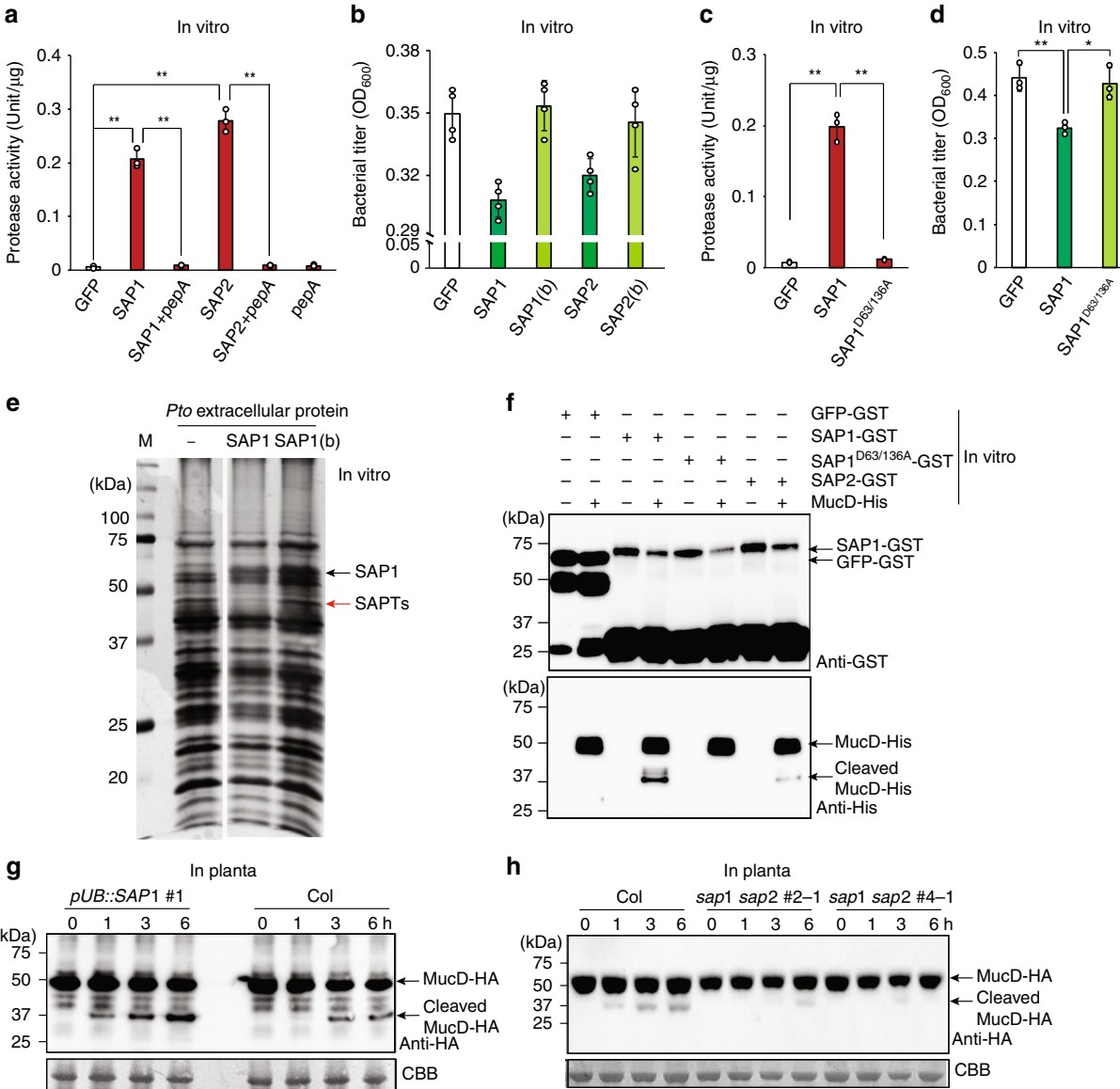

**Fig. 3** Secreted aspartic protease 1 (SAP1) and SAP2 cleave MucD. **a**, **c** Protease activity of purified GFP (green fluorescent protein), SAP1, SAP2 (**a**), and SAP1[D63/136A] (**c**) protein. PepA, the aspartic protease inhibitor, pepstatin A. Data represent means and s.e.m of three biological replicates. Asterisks indicate significant differences (Student's two-tailed $t$ test; **$P < 0.01$). **b**, **d** In vitro growth of $Pto$ ($OD_{600} = 0.05$) in minimal medium supplemented with purified recombinant proteins or boiled (b) recombinant proteins at 9 h after culturing. Data represent means and s.e.m. of four biological replicates. Asterisks indicate significant differences (Student's two-tailed $t$ ; *$P < 0.05$, **$P < 0.01$). **e** Proteins extracted from bacterial culture medium incubated with SAP1, boiled SAP1 (SAP1(b)), or without SAP were visualized (Coomassie Brilliant Blue (CBB) staining). **f** MucD-His incubated with SAP1-GST, SAP1[D63/136A]-GST, and SAP2-GST were visualized by immunoblotting using an anti-GST or anti-His antibody. Three independent experiments were performed with similar results. **g**, **h** $Pto$ $\Delta mucD$ expressing MucD-HA (hemagglutinin) ($OD_{600} = 0.05$) was infiltrated into leaves of 4-week-old Col, $pUB::$ $SAP1-RFP$, and $sap1$ $sap2$ plants. MucD-HA in total protein at the indicated time points was detected by immunoblotting using an anti-HA antibody. CBB staining serves as protein loading control. Experiments were repeated at least three times with similar results

C). Then, we asked whether MucD cleavage is required for SAP-mediated growth suppression of $Pto$. First, we tested whether SAP1 suppresses $Pto$ $\Delta mucD$ MucD[F106Y] growth in vitro. SAP1 but not SAP1[D63/136A] and GFP control suppressed in vitro growth of wild-type $Pto$ and $Pto$ $\Delta mucD$ MucD (Fig. 4e). However, SAP1 did not suppress $Pto$ $\Delta mucD$ MucD[F106Y] growth, indicating that MucD cleavage is essential for SAP1-mediated growth suppression of $Pto$ in vitro (Fig. 4e).

We then tested whether MucD cleavage is required for SAP1-mediated immunity in planta. We infiltrated leaves of Col, $pUB::$ $SAP1-RFP$, $pUB::SAP2-RFP$, $pUB::SAP1\Delta SP-RFP$, $pUB::SAP2\Delta SP-$ $RFP$, and $pUB::SAP1^{D63/136A}-RFP$ plants with wild-type $Pto$, $Pto$ $\Delta mucD$, or $Pto$ $\Delta mucD$ complemented with MucD-HA, MucD[F106Y], or MucD[S394A] and determined in planta bacterial growth. As described above, $Pto$ $\Delta mucD$ growth was compromised and was not different in different host plants (Fig. 4f, g and Supplementary Fig. 5D). In planta growth of wild-type $Pto$ and $Pto$ $\Delta mucD$ MucD-HA was decreased only in plants over-expressing secreted and active SAP1 or SAP2 as compared with wild-type Col plants (Fig. 4f and Supplementary Fig. 5D). Notably, growth of $Pto$ $\Delta mucD$ MucD[F106Y] was not affected by $SAP1$ and $SAP2$ overexpression and was enhanced as compared to wild-type $Pto$, likely because $Pto$ MucD[F106Y] could avoid SAP1- or SAP2-mediated cleavage (Fig. 4f and Supplementary

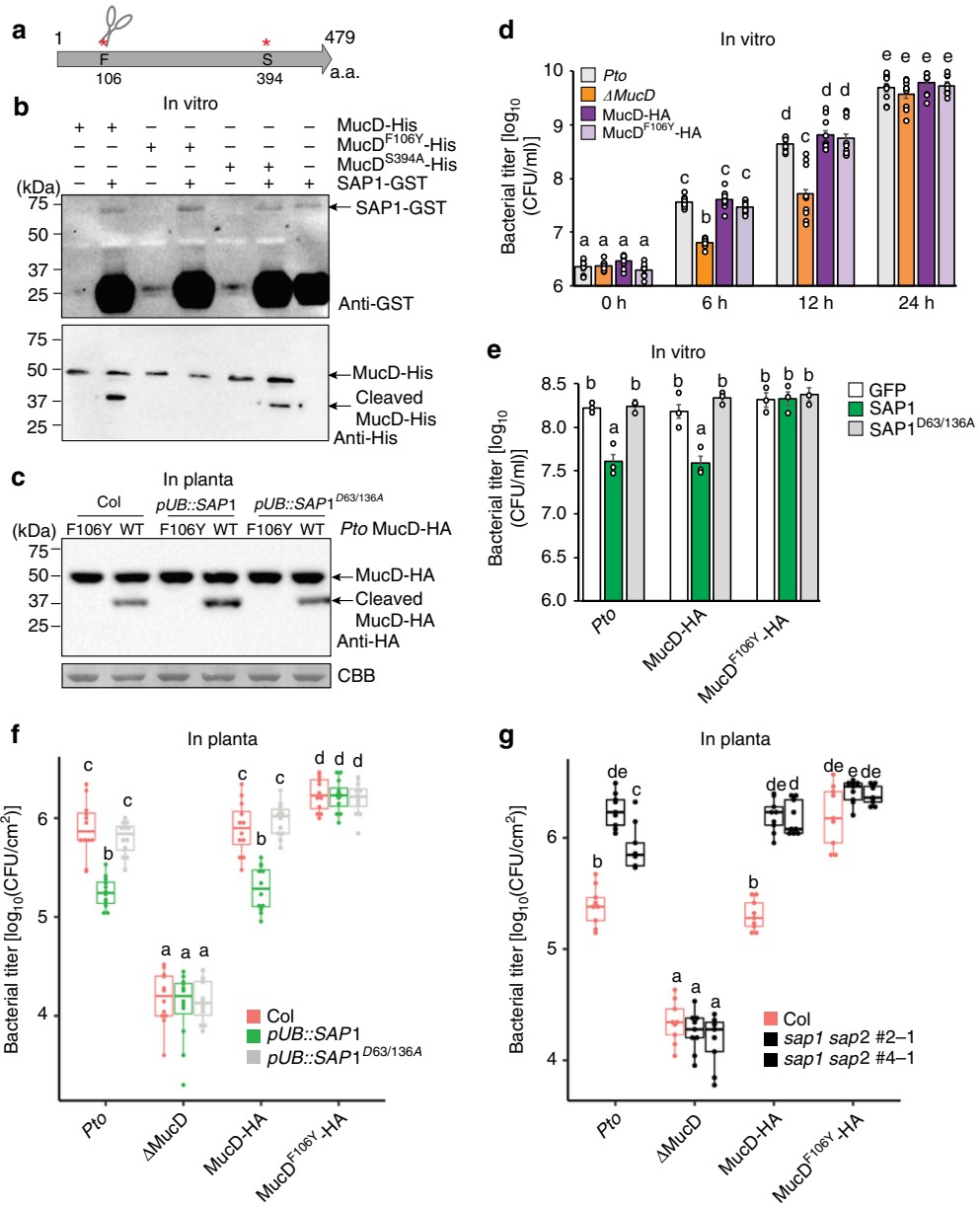

**Fig. 4** Secreted aspartic protease (SAP)-mediated cleavage of MucD suppresses *Pto* growth. **a** Schematic diagram of putative aspartic protease cleavage sites in MucD. The phenylalanine (F) and serine (S) residues were replaced with tyrosine (Y) and alanine (A), respectively. **b** MucD-His, MucD$^{F106Y}$-His, and MucD$^{S394A}$-His proteins were incubated with SAP1-GST. MucD-His or SAP1-GST was visualized by immunoblotting using an anti-HA or anti-GST antibody, respectively. Three independent experiments were performed with similar results. **c** *Pto* Δ*mucD* expressing MucD-HA and MucD$^{F106Y}$-HA (OD$_{600}$ = 0.05) was infiltrated into leaves of 4-week-old Col, *pUB::SAP1-RFP*, and *pUB::SAP1$^{D63/136A}$-RFP* plants. MucD-HA was detected by immunoblotting at 6 hours post infection. Experiments were repeated at least three times with similar results. **d** In vitro growth of *Pto*, *Pto* Δ*mucD*, and *Pto* Δ*mucD* expressing MucD-HA, MucD$^{F106Y}$-HA, or MucD$^{S394A}$-HA (OD$_{600}$ = 0.005) over time. Data represent means and s.e.m. of three independent experiments each with three biological replicates. **e** In vitro growth of *Pto* (OD$_{600}$ = 0.05) supplied with purified recombinant proteins at 9 h after culturing. Data represent means and s.e.m. of three independent experiments with four biological replicates. **f, g** *Pto*, *Pto* Δ*mucD*, and *Pto* Δ*mucD* expressing MucD-HA or MucD$^{F106Y}$-HA were infiltrated into leaves of 4-week-old Col, *pUB::SAP1-RFP*, *pUB::SAP1$^{D63/136A}$-RFP*, and *sap1 sap2* plants, and bacterial titer was determined at 2 days post infection (dpi). Bars represent means and s.e.m. of three independent experiments with at least three biological replicates. **d–g** Statistically significant differences are indicated by different letters (adjusted *P* < 0.01)

Fig. 5D). These results indicate that MucD cleavage was the cause of the enhanced resistance against *Pto* in *SAP*-overexpressing plants. To further demonstrate the physiological significance of SAP-mediated MucD cleavage, we infiltrated leaves of Col and *sap1 sap2* CRISPR-knockout plants with wild-type *Pto*, *Pto* Δ*mucD*, or *Pto* Δ*mucD* complemented with MucD-HA or MucD$^{F106Y}$-HA and determined in planta bacterial growth. As expected, the growth of wild-type *Pto* and *Pto* Δ*mucD* MucD-HA

was enhanced in *sap1 sap2* double mutant plants (Fig. 4g). However, *Pto* Δ*mucD* MucD$^{F106Y}$-HA growth was not affected by *sap1 sap2* mutations (Fig. 4g). Taken together, these results suggest that SAP1 and SAP2 cleave MucD, thereby suppressing *Pto* growth, and that MucD is the major bacterial target of SAP1 and SAP2 for plant immunity.

A previous report showed that the activity of an aspartic protease, CDR1, triggers immune activation in *A. thaliana*[21]. As

overexpression of *SAP1* and *SAP2* did not influence *Pto ΔmucD* MucD$^{F106Y}$-HA growth in planta (Fig. 4f and Supplementary Fig. 5D), activated immune response would be expected to be caused by MucD cleavage. Therefore, we tested whether cleaved fragments of MucD by SAP1 triggers activation of immune responses. We infiltrated leaves of Col plants with wild-type *Pto* or *Pto ΔmucD* MucD$^{F106Y}$-HA and determined expression of the early immune marker gene *FRK1*. To avoid differential immune activation by different bacterial populations, we collected samples at 6 h post infiltration (hpi), where in planta bacterial population of *Pto* and *Pto ΔmucD* MucD$^{F106Y}$-HA were similar (Supplementary Fig. 5E). We observed no significant difference in *FRK1* expression levels between plants infiltrated with wild-type *Pto* or *Pto ΔmucD* MucD$^{F106Y}$-HA (Supplementary Fig. 5F). Thus, MucD cleavage by SAP1 and SAP2 unlikely triggers activation of an immune response effective for bacterial growth suppression.

***mucD* exhibits site-specific diversity in *Pseudomonas*.** Comparative analysis of orthologous gene sequences retrieved from the KEGG database[41] ($n = 2304$) indicates that MucD is highly conserved and widespread in bacteria (Fig. 5a) but not in eukaryotes (Supplementary Data 1). This observation prompted us to test if SAP1 and SAP2 affect the growth of other microbial pathogens in plants. We found that SAP1 and SAP2 restricted the growth of *Pseudomonas cannabina* pv. *alisalensis* ES4326 (*Pca* ES4326; formerly *P. syringae* pv. *maculicola* ES4326), in which MucD is identical to *Pto* MucD at the amino acid level, in a secretion-dependent manner, but did not affect the oomycete pathogen *Albugo laibachii*, which lacks MucD (Fig. 5d, e). Thus, SAP1 and SAP2 may suppress growth of bacteria producing MucD, but have no effect on eukaryotes.

Analysis of selection of bacterial *mucD* sequences using ratios of synonymous to non-synonymous mutations (dN/dS) revealed strong signatures of purifying selection (Fig. 5b). Interestingly, average dN/dS ratio across all sites were significantly higher in sequences from *Pseudomonas* genomes ($n = 92$) compared to the rest of the dataset and to other Gammaproteobacteria ($n = 242$; Fig. 5b). Site-specific analysis of *Pseudomonas* orthologs revealed clusters of residues with higher dN/dS ratio that include F106 whose mutation blocks cleavage by SAP1 and SAP2 (Fig. 5c), indicating that specific regions of *mucD* are under more positive selection in *Pseudomonas*. Positive selection (dN/dS > 1) is difficult to detect probably due to the high frequency of neutral and deleterious mutants. Thus, although dN/dS reaches only ~1, this result might suggest that plant SAPs impose selection pressure on bacterial *mucD*.

## Discussion

During the Past three decades, a large amount of knowledge about how plants recognize microbial molecules and how signal is transduced within plant cells has been acquired. Nevertheless, it is still not understood how plant immunity suppresses the growth of bacterial pathogens that colonize in the apoplast. Here we show that the secreted plant proteases SAP1 and SAP2 serve as a front line of immunity, which inhibits bacterial growth. However, this cleavage of MucD by SAP1 and SAP2 does not appear to result in bacterial death as *Pto ΔmucD* is still viable. This mechanism of partial growth suppression, as opposed to total elimination of bacterial pathogens as is the case in animals[9,42], might explain to some extent the finding that populations of non-virulent plant–bacterial pathogens do not decrease over time[43]. Furthermore, healthy plants in nature are surrounded by diverse bacterial communities that colonize the surface and interior of plant roots and leaves, which do not negatively impact plant fitness[43–47]. Our

findings raise the possibility that secreted plant proteases such as SAP1 and SAP2 might contribute to shaping the plant microbiota and act as a gateway for control of non-virulent and commensal as well as pathogenic bacterial proliferation. It remains to be tested whether SAP-mediated suppression of bacterial growth also occurs in the case of plant endophytes containing MucD from other taxonomic groups and whether the increased positive selection observed in *Pseudomonas* corresponds to an adaptation against SAP-mediated MucD cleavage. Irrespective of this, our work identifies a molecular mechanism by which plant immunity suppresses growth of the bacterial pathogen *Pto*, likely in a direct manner.

Over activation of immunity caused by intraspecific genetic incompatibilities, genetic engineering, or spontaneous mutations increases plant resistance against pathogens but often comes with a growth penalty[6–8]. This so-called immunity-growth tradeoff is shown to be genetically controlled in some cases, which makes production of highly resistant crops that retain growth a difficult task[48,49]. In contrast, overexpression of *SAP* in *A. thaliana* increases resistance against bacterial pathogens, but does not associate with growth retardation or reduced reproduction. Thus, boosting the direct targeting of pathogen factors by plant defense molecules may be an effective strategy to produce disease resistant crops without reducing yield.

Our findings demonstrate that MucD cleavage is the cause of *Pto* growth suppression by SAP1 and SAP2 in *A. thaliana*. MucD from the human opportunistic pathogen *P. aeruginosa* localizes to the periplasm, the space between the inner cytoplasmic membrane and outer membrane, but is also secreted[37,38]. In a human cell line, *P. aeruginosa* MucD is secreted and cleaves interleukin-8, thereby suppressing host immunity[38]. Likewise, *Pto* MucD localizes to the membrane but is also secreted outside the cell. Thus, secreted *Pto* MucD may promote bacterial proliferation in plants and be cleaved by secreted SAP1 and SAP2 in the apoplast. Consistent with this, *Pto* harboring non-cleavable MucD shows higher virulence in plants compared with wild type. Alternatively, secreted SAP1 and SAP2 may also at least partly cleave MucD in the periplasm of *Pto*. Bacteria possess the ability to take up host proteins via membrane transporters[50–52]. Secreted SAP1 and SAP2 may exploit such systems to target MucD. In any case, our finding that both *SAP* and *mucD* are evolutionarily conserved in angiosperms and bacteria, respectively, provides an exciting avenue of research in plant–bacterial interactions.

## Methods

**Plant materials and growth conditions**. *Arabidopsis thaliana* plants were grown in soil in a controlled environment at 22 °C with a 10 h light photoperiod and 65% relative humidity unless otherwise specified. For experiments in sterile conditions, seeds were sterilized with 1.5% sodium hypochlorite and 0.1% Triton X-100 and sown on half strength MS medium (containing half strength MS salts, including vitamins, 1% w/v sucrose, and 0.8% w/v plant agar, pH 5.8) in a controlled environment at 22 °C with a 10 h light photoperiod. All mutants and transgenic plants used in this study were in the background of the *A. thaliana* accession Col. T-DNA insertion lines for *SAP1* (At1g03230; *sap1-1*, SALK_062079 and *sap1-2*, SAIL_646_E08) were obtained from the Nottingham Arabidopsis Stock Center. Homozygous T-DNA insertion mutants were verified by PCR using primers listed in Supplementary Table 2. Accessions *Capsella rubella* (N22697)[53], *Arabidopsis lyrata* (Mn47)[54], *Eutrema salsugineum* (Shandong)[55], and *Solanum lycopersicum* (Moneymaker)[56] were used.

**Transient expression in *A. thaliana***. *Agrobacterium tumefaciens*-mediated transient transformation of *A. thaliana* seedlings was performed as described previously[57]. Briefly, *A. tumefaciens* was cultured in liquid YEB medium at 28 °C to OD$_{600}$ = 1.5, harvested by centrifugation, washed, and resuspended in 0.25× MS pH 6.0, 1% sucrose, 100 μM acetosyringone, and 0.005% Silwet L-77 to OD$_{600}$ = 0.5. The *A. thaliana* seedlings were co-cultivated with *A. tumefaciens* in a 96-well plate in the darkness for 36 h. GFP and RFP fluorescence was detected with an LSM700 confocal microscope (Zeiss Microscopy, Jena, Germany) at standard settings.

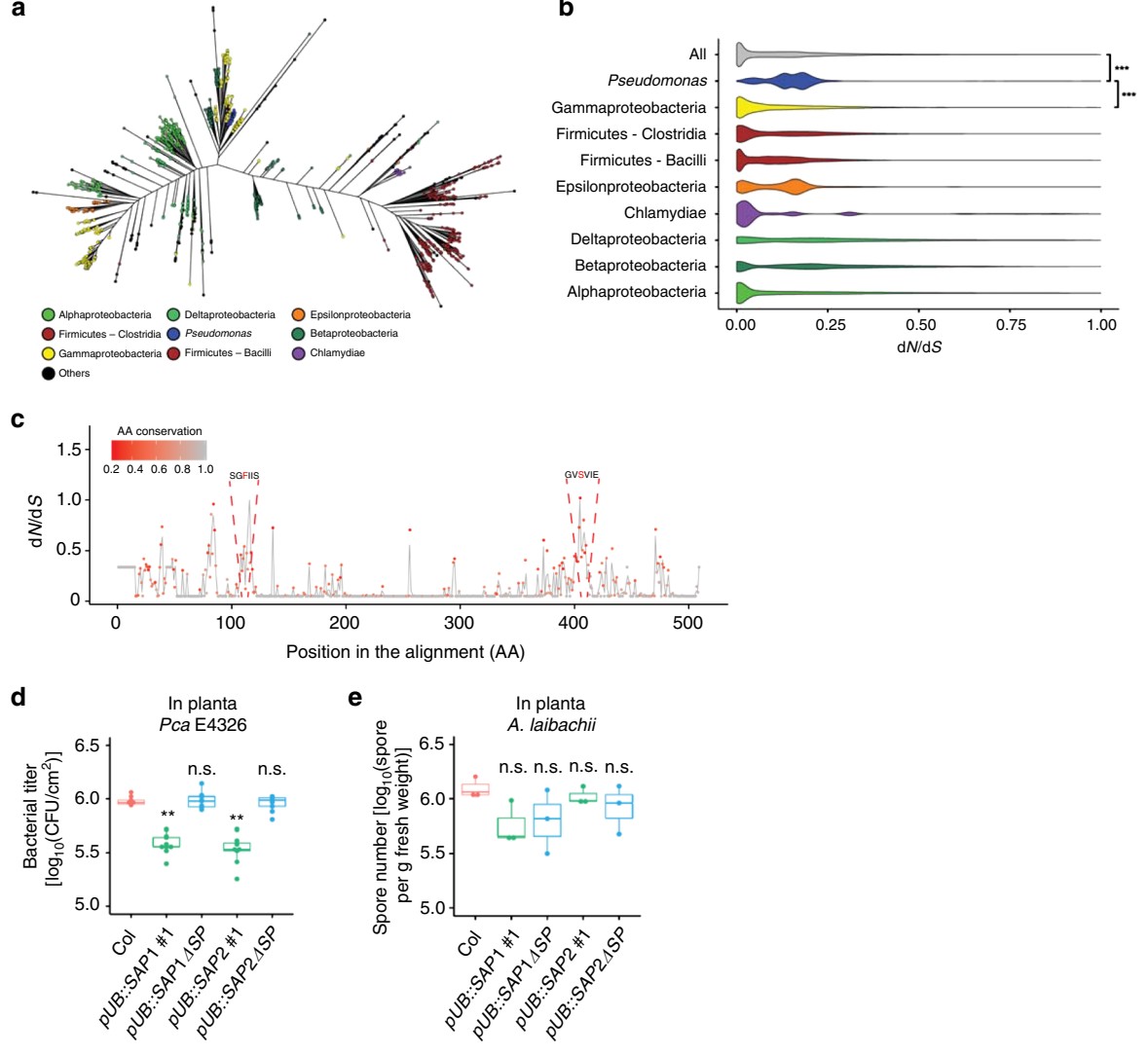

**Fig. 5** MucD is evolutionarily conserved. **a** Maximum-likelihood tree of MucD sequences ($n = 2304$). Colors correspond to the different taxonomic groups. **b** Violin plots showing distributions of pairwise d$N$/d$S$ ratios for each major taxonomic lineage as well as for the entire dataset. Sequences found in the genomes of *Pseudomonas* were removed from the "Gammaproteobacteria" and the "All" groups and were tested for significantly enriched d$N$/d$S$ ratios (Mann–Whitney test; ***$P < 0.001$). **c** Distribution of site-variable d$N$/d$S$ ratios for the multiple sequence alignment of mucD sequences found in *Pseudomonas* genomes ($n = 92$). The color scale corresponds to the amino acid conservation rate at each site. Putative aspartic protease cleavage sites are indicated in red. **d** *Pca* E4326 ($OD_{600} = 0.001$) was infiltrated into leaves of 4-week-old Col, *pUB::SAP1-RFP* and *pUB::SAP2-RFP* plants, and bacterial titer was determined at 2 dpi. Bars represent means and s.e.m of three independent experiments with three biological replicates. Asterisks indicate significant differences from Col (Student's two-tailed *t* test; **$P < 0.01$). **e** Col, *pUB::SAP1-RFP*, and *pUB::SAP2-RFP* were infected with *Albugo laibachii*, and the number of spores was measured at 14 dpi. Bars represent means and s.e.m. of three independent biological replicates

**Preparation of inocula and bacterial growth assay**. *Pto* strains[58] were grown overnight in King's B medium supplemented with 50 μg/ml of rifampicin and *P. cannabina* pv. *alisalensis* ES4326 was grown overnight in King's B medium supplemented with 50 μg/ml of streptomycin. The bacteria were harvested by centrifugation, washed, and diluted to the desired density with sterile water. The bacterial growth assay was performed as described before[59]. Briefly, 4- to 5-week old leaves were syringe inoculated with bacterial suspension using a needleless syringe. A leaf disc collected from the infiltrated leaf was considered a biological replicate. In each experiment, six different plants were infiltrated.

**RNA isolation and RT-qPCR**. Total RNA was isolated from plant samples using TRIzol reagent (Thermo Fisher Scientific) following the manufacturer's instructions. Five micrograms of total RNA were reverse transcribed using the SuperScript II First-Strand Synthesis System (Thermo Fisher Scientific) with an oligo(dT) primer. Real-time DNA amplification was monitored using Bio-Rad iQ5 optical system software (Bio-Rad). The expression level of genes of interest was normalized to that of the endogenous reference gene *ACTIN2*. The used primers are listed in Supplementary Table 2.

**Extraction of apoplastic fluid**. Apoplastic fluid was extracted from 4-week-old *A. thaliana* leaves at 24 h post treatment with water or 1 μM of flg22. Leaves were collected and washed with sterilized water for two times. Leaves were submerged in sterilized water, vacuum infiltrated for 10 min, and released gently. Waters attached on leaves were carefully removed, and apoplastic fluid was collected after centrifuge at $1000 \times g$ for 10 min at 4 °C. The extracted apoplastic fluid was filtered through a 0.22-μm filter.

**Total and apoplastic protein extraction**. Total and apoplastic protein was extracted from *A. thaliana* leaves as described previously[60]. Briefly, for total protein extraction, *A. thaliana* leaves were frozen in liquid nitrogen and ground to a fine powder. Protein extraction buffer (0.5 M Tris-HCl, pH 8.3, 2% v/v Nonidet P-40, 20 mM $MgCl_2$, 2% v/v β-mercaptoethanol, 1 mM phenylmethylsulfonyl fluoride, and 1% w/v polyvinyl polypyrrolidine) and Tris-saturated phenol (pH 7.9) were added to the tissue powder, and then mixed gently at room temperature for 10 min. After centrifugation at $12,000 \times g$ for 15 min at 4 °C, the phenol layer was transferred into a new tube and precipitated with methanol containing 100 mM ammonium acetate for 2 h at −20 °C. The precipitated protein pellet was washed twice with methanol containing 100 mM ammonium acetate and twice with 80%

acetone. For apoplastic protein extraction, *A. thaliana* leaves were shaken in apoplastic extraction buffer (200 mM CaCl₂, 5 mM sodium acetate pH 4.3, and protease inhibitor cocktail) on ice for 1 h. The extraction buffer was filtered through No. 2 filter paper (Whatman, Cambridge, UK) and mixed thoroughly after adding a half volume of Tris-saturated phenol. The phenolic phase was transferred into a new tube after centrifugation at $5000 \times g$ for 15 min. Protein was precipitated and washed as described above.

**Immunoblotting**. Equal amounts of protein were separated by 12% sodium dodecyl sulfate-polyacrylamide gel electrophoresis (SDS-PAGE) and transferred to PVDF membranes (Sigma-Aldrich, St. Louis, MO, USA) with Mini-PROTEAN® Tetra Handcast Systems (Bio-Rad, Hercules, CA, USA). The membranes were incubated with primary antibodies (for anti-GFP (Abcam, AB6556), 1:5000; anti-HA (Roche, 11867423001), 1:5000; anti-PR1, 1:5000; anti-RbCL (Clontech 632475), 1:2000; anti-GST (Virogen, 101-A-100), 1:5000; or anti-His (Thermo Fisher, PA1-23024), 1:5000) in 1× TBS with 5% w/v skim milk at 4 °C overnight. Treated membranes were washed with 1× TBS and incubated with a secondary horse radish peroxidase (HRP)-conjugated antibody (anti-rabbit HRP (Santa Cruz, sc-2004), 1:10,000; anti-rat (Santa Cruz, sc-2006), 1:10,000; anti-mouse (GE Healthcare Life Science; NA931), 1:10,000) at room temperature for 2 h. Signals were visualized using the Pierce ECL and ChemiDoc MP (Bio-Rad) systems. Protein band intensity was measured by using ImageJ software. Relative intensity protein band under *Pto* infection condition was calculated by normalizing to the intensity of protein band under mock condition, which was set as 1.

**Construction of transgenic plants**. The coding or promoter sequences of *SAP1* (At1g03230) and *SAP2* (At1g03220) were PCR amplified using the primers listed in Supplementary Table 2, cloned into pENTR/D-TOPO (Thermo Fisher Scientific, Carlsbad, CA, USA), and transferred into pUB::C-RFP:GW or pFAST-G04 vectors[61], respectively. To generate the *SAP2*-RNAi line, a 547 bp fragment was amplified from the *SAP2* coding sequence using the primers listed in Supplementary Table 2, cloned into the pENTR/D-TOPO vector, and transferred into the pFAST-G03 vector[61]. *Agrobacterium tumefaciens* strain GV3101/pMP90 was transformed with each plasmid. The resultant agrobacteria were used to generate stable *A. thaliana* transgenic plants using the floral dip method (Clough and Bent, 1998). Transgenic plants were selected by spraying with BASTA (pUB::C-RFP:GW) or germinating on MS medium containing kanamycin (pFAST-G04) or hygromycin (pFAST-G03). The *SAP2 CRISPR-CAS* mutants were generated in Col and *sap1-1* backgrounds with the CRISPR-Cas9 system from the Karlsruhe Institute of Technology following the provided protocol[62–64].

**Identification of candidate SAP1 target proteins by LC-MS/MS**. Protein bands in silver-stained SDS-PAGE were excised with blades, washed with 50% v/v acetonitrile in 0.1 M NH₄HCO₃, and dried in a vacuum centrifuge. Gel fragments were treated with a reducing buffer (10 mM DTT, 0.1 M NH₄HCO₃) for 45 min at 55 °C. After removal of the reducing buffer, 55 mM iodoacetamide in 0.1 M NH₄HCO₃ was added. Gel pieces were then dried, submerged in digestion buffer (25 mM NH₄HCO₃, and 12.5 ng/ml trypsin), and incubated at 37 °C overnight. Tryptic peptides were analyzed by MS as previously described[65].

**Recombinant protein expression and protease activity assay**. *GFP*, *SAP1*, and site-mutagenized *SAP1* and *SAP2* were cloned into pDEST15 (Thermo Fisher Scientific), and *mucD* was cloned into pDEST59 (Invitrogen) for recombinant protein expression. The recombinant proteins were expressed in *E. coli* and purified through affinity pull-down with Pierce Glutathione agarose beads (Thermo Fisher Scientific) or Ni-NTA agarose beads (Qiagen, Hilden, Germany) according to the supplier's instructions. The universal protease activity assay was carried out using purified recombinant protein as described previously[66]. Briefly, recombinant protein was added to a 0.65% w/v casein solution, and followed by incubation at 37 °C for 10 min. Trichloroacetic acid solution (110 mM) was added to stop the reaction, and the solution was incubated at 37 °C for 30 min. The supernatant was collected by filtration through a 0.45 μm polyethersulfone syringe filter. Folin and Ciocalteu's phenol reagent (Sigma-Aldrich) and 500 mM sodium carbonate solution were added to the filtered solution (1:2 v/v and 2.5:1 v/v), respectively. The samples were then mixed and incubated at 37 °C for 30 min. The solution was collected after centrifugation at $3,000 \times g$ for 10 min, and absorbance was measured by a spectrophotometer at 660 nm. Enzyme activity was calculated based on a standard curve using L-tyrosine as the standard. Pepstatin A (Sigma-Aldrich) was used as an aspartic protease inhibitor. Pepstatin A was pre-incubated with the purified recombinant protein at a concentration of 1 μM for 10 min prior to the protease activity assay.

**Generation of bacterial mutant and complementation lines**. A *Pto mucD* (PSPTO_4221) gene deletion mutant was created as previously described[67]. The upstream and downstream adjacent regions of *mucD* and a gentamycin resistance gene were amplified and linked together by PCR. This PCR product was then digested with *Bam*HI and *Hin*dIII and cloned into the MCS of pK18mobsacB[67]. The plasmid was then used to generate Δ*mucD* by a triparental mating using the helper plasmid pRK2013. The *mucD* coding sequence was amplified from *Pto*

genomic DNA by PCR and cloned into the pENTR/D-TOPO vector, and then transferred to pCPP5040 by LR reaction. The complementation strains generated by a triparental mating of *E. coli* carrying pCPP5040::MucD-HA, pCPP5040:: MucD^F106Y-HA, or pCPP5040::MucD^S394A-HA with the *mucD* deletion strain and a strain carrying pRK2013 and were selected with 50 μg/ml rifampicin, 5 μg/ml gentamycin, and 35 μg/ml chloramphenicol.

**Bacterial membrane and secreted protein extraction**. *Pto* MucD-HA was grown in King's B medium overnight, washed with water twice, inoculated into M9 minimal medium (start OD₆₀₀ = 0.05), and cultured at 28 °C for 6 h. After centrifugation at $6000 \times g$ for 10 min, bacterial cells and culture medium were used for membrane protein and secreted protein extraction, respectively. A carbonate extraction method was used for bacterial cell membrane isolation[68]. Briefly, the cell pellet was washed with wash solution (50 mM Tris-HCl, pH 7.5) and centrifuged at $2500 \times g$ for 8 min. The pellet was then resuspended in wash solution containing DNase I. The cells were ruptured by sonication, and unbroken cells were removed by centrifugation at $2500 \times g$ for 8 min. The supernatant was directly added to icecold 100 mM sodium carbonate solution, and gently stirred on ice for 1 h. The cell membranes were collected by ultracentrifugation at $115,000 \times g$ for 1 h at 4 °C. The membrane pellet was resuspended in the wash solution, and collected by ultracentrifugation. Secreted protein in the culture medium was extracted with the phenol method[69]. Briefly, the culture medium was centrifuged again at $12,000 \times g$ for 10 min, and the supernatant was collected and mixed with Tris-saturated phenol. After thorough mixing, the phenolic layer was collected by centrifugation at $3000 \times g$ for 10 min. Protein was precipitated and washed as described above.

**Evolutionary analysis of bacterial *mucD* sequences**. We first retrieved all *mucD*-orthologous sequences found in the bacterial genomes present in the KEGG Ortholog (KO) database[41] (accessed on 15/10/2016). Next, we performed a multiple sequence alignment at the amino acid level using Clustal Omega[70]. Nucleotide sequences were then aligned by codon using Pal2Nal[71]. Based on this multiple sequence alignment, a species tree was inferred using FastTree[72]. We then employed the PAML software[73] to obtain, for each taxonomic group, pairwise dN/dS ratios using the M0 model and separately for sequences retrieved from *Pseudomonas* genomes using the M8 model, which allows dN/dS ratios to vary among sites. Distributions of pairwise dN/dS ratios were compared using the non-parametric Mann–Whitney test. *P* values were corrected for multiple testing using the Bonferroni method, with a significance threshold of $\alpha = 0.05$.

**Albugo preparation and infection**. *Albugo laibachii* Nc14 spores from infected Col plants were collected from leaf washes and treated on ice for 30 min to release zoospores. Zoospores were collected by filtration and sprayed on *A. thaliana* leaves at a concentration of $5 \times 10^4$ conidiospores/ml. The inoculated plants were maintained in a growth chamber at 22 °C 16 h day/16 °C 8 h night with 100% relative humidity, and then moved to 75% relative humidity conditions after 36 h. The number of released spores on leaves were counted at 14 days post infiltration (dpi) as described previously[74].

**Phylogenetic analysis of SAP1 homologs**. The entire protein sequences of *A. thaliana, A. lyrata, C. rubella, C. grandiflora, E. salsugineum, B. rapa*, tomato, and rice were retrieved from Phytozome[75] and used for identification of putative orthologous groups using the OrthoMCL program[76]. The used genes were *CrubASP1* (Crubv10011636m), *CrubASP2* (Crubv10025694m) *AlyrASP1* (Alyr484681), *AlyrASP2* (Alyr470302), *EsalASP* (Esalv10007672), *SolycASP* (Solyc01g080010.2.1), *OsSAPa* (Os05g33400.1), *OsSAPb* (Os05g33410.1), and *OsSAPc* (Os05g33430.1). The proteins belonging to the same group as *A. thaliana* SAP1 and SAP2 were aligned using MUSCLE[77]. A maximum likelihood phylogenetic tree was constructed using the MEGA6 software[78].

**Statistical analysis**. The following models were fit to the relative cycle threshold (Ct) values compared to *Actin2* (for qRT-PCR) or log₁₀-transformed bacterial titers (for bacterial titer) with the lmer function in the lme4 package or the lm function in the R environment: $C_{tgytr} = G \Upsilon T_{gyt} + R_r + \varepsilon_{gytr}$, where $G \Upsilon T$ is the genotype–treatment–time interaction, and random factors; $R$ the biological replicate; $\varepsilon$ the residual; $C_{tgyr} = G \Upsilon_{gy} + R_r + \varepsilon_{gytr}$, where $G \Upsilon$ is the genotype–treatment interaction; $C_{tgtr} = G T_{gt} + R_r + \varepsilon_{gtr}$, where $G T$ is the genotype–time interaction. The mean estimates of the fixed factors were used as the modeled relative Ct values visualized as the relative log₂ expression values or bacterial titers. Differences between estimated means were compared using two-tailed *t* tests. For the *t* tests, the standard errors appropriate for the comparisons were calculated with the variance and covariance values obtained from the model fittings. The Benjamini–Hochberg method was applied to correct for multiple hypothesis testing when all pairwise comparisons of the mean estimates were made.

**Reporting summary**. Further information on research design is available in the Nature Research Reporting Summary linked to this article.

## Data availability

The source data underlying Figs. 1a–c, 2a–c, 2e–i, 3a–h, 4b–g, 5d, e and Supplementary Figs. 1D, F, 2C-E, 3C, 4A-D, 5B-F are provided as a Source Data file.

## Code availability

All R codes used for statistical analyses are available from the authors.

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

## Acknowledgements

We thank Xinnian Dong for kindly providing the anti-PR1 antibody, Alan Collmer for pK18mobsacB and pCPP5040, Neysan Donnelly for scientific English editing, and Fumiaki Katagiri, Jane Glazebrook, Imre Somssich, Jane Parker, and Paul Schulze-Lefert for critical reading of the manuscript. Y.W. was a recipient of a Postdoctoral Fellowship from Alexander von Humboldt-Foundation and Bayer Science and Education Foundation. This work was supported by the Max Planck Society (K.T.).

## Author contributions

Y.W. carried out most experiments, analyzed data, and wrote the paper; J.W. performed the in vitro bacterial growth assay; T.M.W. performed expression analysis of *SAP* homologs; M.A. and E.K. performed the oomycete infection assay; R.G.-O. carried out the evolutionary analysis of *mucD*; T.C. performed MS analysis; T.N. analyzed data. K.T. conceived and coordinated the research, analyzed data, and wrote the paper.

## Additional information

**Competing interests:** The authors declare no competing interests.

