## [Peer Review File · Nature Communications]

Reviewers' comments:

Reviewer #1 (Remarks to the Author):

The authors show that *Arabidopsis thaliana* Secreted Aspartic Protease 1 and 2 (SAP1 and SAP2) cleave *Pseudomonas syringae* pv. tomato DC3000 (Pto) protease MucD causing inhibition of the growth of the bacterial pathogen on *A. thaliana*. The authors then perform all the experiments that are required to prove that SAP1 and SAP2 cleave MucD, which is supposed to be a virulence factor. Replacing MucD proteolytic cleavage sites F106 by Y and S394 by A prevented cleavage by SAP1 and SAP2. MucDF106Y was still functional and could complement the MucD deletion mutant, but was no longer cleaved by SAP1 and SAP2. MucDS394 A could not complement the MucD deletion mutant. The cleavage products of MucD did not induce local or systemic resistance as reported for aspartic protease CDR1 published before. Both SAP1 and SAP2 as well as MucD do occur in many different plant species and bacteria, respectively.

However, the virulence function of MucD has shown before in *Pseudomonas aeruginosa* that is both an animal and plant pathogen. The novelty of the paper is that SAP1 and SAP2 suppress bacterial growth which seems mainly to be due to cleaving virulence factor MucD. The exact cleavage products of MucD not known yet. Overexpression of SAP1 and SAP2 did not have a immunity growth tradeoff in *A. thaliana*. All experiments seem to be carried out in a solid way.

The main problems I have with the paper is the levels of bacterial growth inhibition found on which the conclusions are based. These are often less than half a log unit. In the original paper on MucD by Yorgey et al. (2001; Mol Microbiol 41, 1063-1076) log units of two were reported for MucD deletion mutants. Also different bacterial growth titers in infected plants were reported. Also different inoculum concentrations were used to inoculate plants (OD varied from 0,001 to 0,05). Different concentrations were used in vitro and in planta. Most worrying are the very small differences in log CFU units (sometimes smaller than 0,3 units. Are they significant? Another question remains. Suppose that MucD is a virulence factor with a target in the apoplast, why would SAP1 and SAP2 have the same effects in vitro when this virulence factor is not needed? Another question that remains is whether MucD a (serine) protease or does it have an additional function (chaperone?). In *Pseudomonas aeruginosa* it was suggested it could be involved in generating a toxin (in case of killing *C. elegans*). In conclusion all experiments have performed very well, but I wonder whether the inoculation results warrant the conclusions drawn. The statistics could be explained in more detail.

Reviewer #2 (Remarks to the Author):

Wang et al report that two *Arabidopsis* aspartic proteases, ASP1 and ASP2, are secreted to the extracellular space to cleave the bacterial MucD protein as an antibacterial mechanism. The mutation of ASP1 and ASP2 led to compromised disease resistance whereas the mutation of MucD reduces pathogen's growth in planta. Although the exact function of MucD is unclear, the work revealed a defense/pathogenesis mechanism that was previously unknown.

Main concerns:

1. The results would be more convincing if the phenotypes of the *sap1 sap2* double mutants could be complemented at least by a SAP1 transgene (since the double mutants was based on a single *sap1* mutant line). Did the authors also determine whether the mutant phenotypes could be complemented by the inactive variants (such as D63/136A)?
2. The protease activity of both SAP1 and SAP2 on MucD was apparently very weak. For instance, it appears that only about 10% of MucD was cleaved by ASP in the in-vitro enzymatic assay (Fig 3E). If the enhanced resistance mediated by SAP1 and SAP2 is not due to activation of the host immune response by a cleavage product of MucD, how to explain that the pathogen's growth could still be strongly inhibited by ASP1 and ASP2 even if a large majority of MucD remain intact?

Minor concerns:

3. In lines 131-132, the authors stated that none of transgenic plants showed growth retardation, or reduced reproduction (Figures 2E-2F). The statement was not accurate as significant increase or decrease in fresh weight or silique numbers was observed in the lines expressing UB::SAP1, UB::SAP2, or the truncated SAP1 and SAP2 based on the data (Fig 2E-2F). Besides, UB:SAP2#2 also showed increased PR1 expression.
4. The authors stated that a 50 kDa band from the bacteria was reduced by ASP1 (Fig 3D). However, based on Fig 3D, the protein (the SAPTs) is clearly much smaller than 50 kDa, perhaps close to 45 kDa. It would be more convincing if the authors could do a similar experiment by using the mucD mutant to verify that the band putatively cleaved by SAP1 is indeed MucD.
5. Regarding the cleavage sites of MucD by SAP1 and SAP2, the authors purified these proteins and did the cleavage assay and then detected the cleavage products by Western analysis. Using mass spec assay to tell the cleavage sites of MucD would be more informative than the Western blot analysis.
6. Fig 3E: What is the 28 kDa band detected by anti-GST?
7. Lines 198-199: the authors stated that "We found that Pto MucD was localized to the membrane and was secreted outside of the cell (Figure S5B)". The statement seems contradictory. Localized to the membrane or secreted outside of the cell?
8. Different sizes of MucD and its cleavage products were shown in the different figures. In Fig 3E, MucD-His was much smaller than 50 kDa and the cleaved MucD-His was about 30 kDa. However, in the other figures, MucD was about 50 kDa and the cleavage product was 37 kDa (such as in Fig 4B).

Reviewer #3 (Remarks to the Author):

This manuscript reports a well-performed series of experiments that convincingly show that site-specific cleavage of MucD by two homologous secreted aspartic proteases increases immunity against bacteria. I like this manuscript because it includes a large amount of carefully performed experiments that clearly demonstrate the direct link between a host protease and pathogen substrate. Especially the use of a non-cleavable substrate mutant, showing that this enhances virulence irrespective of the protease is neat. This story is well-suited for this journal.

I do, however, have two minor main points and several minor points:

- 1) Fig1A shows OD, but it is unclear if the higher OD is caused by stuff accumulating in the apoplast of flg22-treated plants or caused bacterial growth. Please show the number of life bacteria, as used in the rest of the manuscript, or show a $t=0$.
- 2) I am puzzled by the quick selection of two putative cleavage sites for an uncharacterized protease. Please explain more carefully how these sites were selected.

Minor points:

- 1) I think it should be mentioned that MucD is a virulence factor in both title and abstract. The way it is written now is neutral.
- 2) L107 write 'Immunoblotting of the SAP1-GFP fusion...'
- 3) FigS2D should be a main figure.
- 4) Explain better why MucD was selected from 21 SAPTs
- 5) L181 'SAP1 and SAP2 are required to cleave'

- 6) L308 states that 'we did not address where SAP1 and SAP2 cleave MucD'. What do you mean? Is it not better to state that you have not 'confirmed that cleavage occurs in the apoplast'?
- 7) M&M L370 states that apoplastic extraction buffer contains a protease inhibitor cocktail. Would that not interfere with protease assays done throughout the manuscript?
- 8) Would it be possible to add stats to Fig1D (confocal images?)
- 9) Why was Fig2H not part of Fig2B, like Fig2E and 2F?
- 10) Why was Fig3A not combined with Fig3C? SAP1 seems identical between the two.
- 11) Why were the bacterial growth assays done with boiled SAP1 and not with the SAP1 mutant or pepstatin?
- 12) The readability of the figures would improve if it stated *in vitro*/*in vivo* or some other info.
- 13) Fig5C is printed too small to read.

Point-to-point response letter

Reviewer #1 (Remarks to the Author):

The authors show that *Arabidopsis thaliana* Secreted Aspartic Protease 1 and 2 (SAP1 and SAP2) cleave *Pseudomonas syringae* pv. tomato DC3000 (Pto) protease MucD causing inhibition of the growth of the bacterial pathogen on *A. thaliana*. The authors then perform all the experiments that are required to prove that SAP1 and SAP2 cleave MucD, which is supposed to be a virulence factor. Replacing MucD proteolytic cleavage sites F106 by Y and S394 by A prevented cleavage by SAP1 and SAP2. MucDF106Y was still functional and could complement the MucD deletion mutant, but was no longer cleaved by SAP1 and SAP2. MucDS394 A could not complement the MucD deletion mutant. The cleavage products of MucD did not induce local or systemic resistance as reported for aspartic protease CDR1 published before. Both SAP1 and SAP2 as well as MucD do occur in many different plant species and bacteria, respectively.

However, the virulence function of MucD has shown before in *Pseudomonas aeruginosa* that is both an animal and plant pathogen. The novelty of the paper is that SAP1 and SAP2 suppress bacterial growth which seems mainly to be due to cleaving virulence factor MucD. The exact cleavage products of MucD not known yet. Overexpression of SAP1 and SAP2 did not have a immunity growth tradeoff in *A. thaliana*. All experiments seem to be carried out in a solid way.

The main problems I have with the paper is the levels of bacterial growth inhibition found on which the conclusions are based. These are often less than half a log unit. In the original paper on MucD by Yorgey et al. (2001; Mol Microbiol 41, 1063-1076) log units of two were reported for MucD deletion mutants. Also different bacterial growth titers in infected plants were reported.

(Response 1)

We observed that *mucD* deletion caused compromised *Pseudomonas syringae* pv. *tomato* DC3000 growth in plants (one to two log₁₀ unit; Figure 4F and 4G), which was not very different from Yorgey et al. Please also note that we used *Pseudomonas syringae* in this study and Yorgey et al used *Pseudomonas aeruginosa*. Thus, the consequence of *mucD* deletion is similar in *P. syringae* and *P. aeruginosa* (loss of virulence) with varied degree, likely due to different bacterial species used.

Also different inoculum concentrations were used to inoculate plants (OD varied from 0,001 to 0,05). Different concentrations were used in vitro and in planta. Most worrying are the very small differences in log CFU units (sometimes smaller than 0,3 units. Are they significant?

(Response 2)

We used different doses of bacteria *in planta* assays, depending on the purpose of experiments, which is the norm in the field. For instance, lower OD was used for bacterial growth assay in plants. Higher OD was used to detect the cleavage of MucD protein and gene expression changes. Although bacterial growth in *sap1 sap2*-RNAi lines showed weak phenotypes ($\sim 0.5 \log_{10}$ unit), they were statistically different from wild-type plants (Fig S1F). To confirm our data, we generated *sap1 sap2* double mutants with the CRISPR/Cas9 system. We observed 0.5 to 1 \log_{10} differences in bacterial growth for important comparisons. For instance, the difference between wild-type and *sap1 sap2* double mutants showed around 1 \log_{10} differences, which is a 10 fold difference (Figure 2A). We presented statistical significances of all relevant comparisons with asterisks or letters.

Another question remains. Suppose that MucD is a virulence factor with a target in the apoplast, why would SAP1 and SAP2 have the same effects *in vitro* when this virulence factor is not needed?

(Response 3)

mucD was required for growth not only *in planta* but also *in vitro* while the effect of *mucD* deletion was larger in plants. Thus, MucD is likely a virulence factor in plants but is also required for general growth.

Another question that remains is whether MucD a (serine) protease or does it have an additional function (chaperone?). In *Pseudomonas aeruginosa* it was suggested it could be involved in generating a toxin (in case of killing *C. elegans*). In conclusion all experiments have performed very well, but I wonder whether the inoculation results warrant the conclusions drawn. The statistics could be explained in more detail.

(Response 4)

We agree that the question how MucD functions in plants is interesting, but we think that it is beyond our scope of this manuscript. We hope to address this question in the future. We performed statistical analyses with linear mixed models followed by two-tailed t-tests. When necessary, we performed multiple test corrections by the Benjamini-Hochberg method. We described these detailed statistical analyses in Method section. We hope that this is sufficient.

Reviewer #2 (Remarks to the Author):

Wang et al report that two Arabidopsis aspartic proteases, ASP1 and ASP2, are secreted to the extracellular space to cleave the bacterial MucD protein as an antibacterial mechanism. The mutation of ASP1 and ASP2 led to compromised disease resistance whereas the mutation of MucD reduces pathogen's growth in planta. Although the exact function of MucD is unclear, the work revealed a defense/pathogenesis mechanism that was previously unknown.

Main concerns:

1. The results would be more convincing if the phenotypes of the *sap1 sap2* double mutants could be complemented at least by a SAP1 transgene (since the double mutants was based on a single *sap1* mutant line). Did the authors also determine whether the mutant phenotypes could be complemented by the inactive variants (such as D63/136A)?

(Response 5)

Thank you for the critical comment. However, we also generated multiple RNAi lines (*sap1 sap2*-RNAi), which showed susceptible phenotypes (Figure S1F). *sap1* single mutants did not show bacterial growth phenotypes, so susceptible phenotypes of double mutants should be caused by CRISPR mutations or RNA silencing in *SAP2*. In addition, secretion-dependent effects of overexpression of *SAP1* and *SAP2* (Figure 2B) and no enhanced growth of *Pto ΔmucD MucD^{F106Y}* in *sap1 sap2* mutants as compared to wild type plants (Figure 4G) support that susceptible phenotypes of double mutants were caused by mutations in *SAP1* and *SAP2*. We did not complement *sap1 sap2* mutants with an inactive variant of *SAP1*. However, as plants overexpressing *SAP1* D63/136A showed a wild type phenotype for bacterial growth (Figure 2H), we expect that *SAP1* D63/136A does not complement *sap1 sap2* mutants.

2. The protease activity of both *SAP1* and *SAP2* on MucD was apparently very weak. For instance, it appears that only about 10% of MucD was cleaved by ASP in the in-vitro enzymatic assay (Fig 3E). If the enhanced resistance mediated by *SAP1* and *SAP2* is not due to activation of the host immune response by a cleavage product of MucD, how to explain that the pathogen's growth could still be strongly inhibited by *ASP1* and *ASP2* even if a large majority of MucD remain intact?

(Response 6)

We agree that not all MucD protein was cleaved by *SAPs* *in vitro* and *in planta*. This is consistent with our observation that wild type *Pto* in *SAP* overexpression plants grew more than *Pto ΔmucD* in wild type plants (Figure 4F). In addition, effects of *SAPs* *in vitro* and *in planta* were dependent on MucD cleavage as *Pto ΔmucD MucD^{F106Y}* grew similarly in wild type and *SAP* overexpression plants (Figure 4F) and wild type and *sap1 sap2* mutants (Figure 4G). Thus, even though MucD cleavage by *SAPs* is incomplete, the partial MucD cleavage has significant effects on bacterial growth.

Further investigation will be required to answer how incomplete cleavage of MucD affects bacterial growth *in vitro* and *in planta*, which is a future issue.

Minor concerns:

3. In lines 131-132, the authors stated that none of transgenic plants showed growth retardation, or reduced reproduction (Figures 2E-2F). The statement was not accurate as significant increase or decrease in fresh weight or silique numbers was observed in the lines expressing UB::SAP1, UB::SAP2, or the truncated SAP1 and SAP2 based on the data (Fig 2E-2F). Besides, UB:SAP2#2 also showed increased PR1 expression.

(Response 7)

We apologize our oversight. We have changed the text accordingly. Now it reads

“Interestingly, none of transgenic plants showed enhanced expression of the immune marker *PR1*, except for *pUB::SAP2* line 2 with a slight increase, growth retardation, or reduced reproduction, but some of them showed enhanced growth and reproduction (Fig. 2c-f).”

4. The authors stated that a 50 kDa band from the bacteria was reduced by ASP1 (Fig 3D). However, based on Fig 3D, the protein (the SAPTs) is clearly much smaller than 50 kDa, perhaps close to 45 kDa. It would be more convincing if the authors could do a similar experiment by using the mucD mutant to verify that the band putatively cleaved by SAP1 is indeed MucD.

(Response 8)

The band contained other bacterial proteins in addition to MucD as we detected other bacterial proteins in the MS analysis. Thus, we did not claim that MucD was the sole protein in the band. We apologize mistakes introduced when we prepared immunoblotting figures. Indications of protein size markers were misaligned. We have corrected this in relevant figures. Migration of MucD appears slightly slower than its predicted protein size, which might be caused by its amino acid compositions and/or possible protein modifications. In addition, we showed that MucD but not MucD F106Y could be cleaved by SAPs *in vitro* and *in planta* (Figure 3, Figure S2, and Figure S4). We think that these data are sufficient to demonstrate that SAPs cleave MucD *in vitro* and *in planta*.

5. Regarding the cleavage sites of MucD by SAP1 and SAP2, the authors purified these proteins and did the cleavage assay and then detected the cleavage products by Western analysis. Using mass spec assay to tell the cleavage sites of MucD would be more informative than the Western blot analysis.

(Response 9)

MucD F106Y was not cleaved by SAP1 both *in vitro* and *in planta*, suggesting that SAP1 cleaves MucD likely at F106 position. While we agree that defining the exact cleavage site by mass spec would be informative, we think that this is beyond the scope of our current manuscript.

6. Fig 3E: What is the 28 kDa band detected by anti-GST?

(Response 10)

As this band was commonly detected in different GST fusion proteins (GFP, SAP1, and SAP2), we predict that the band comes from an *E. coli* protein, which was co-purified GST fusion proteins and cross-reacted with the GST antibody, or a degraded GST product.

7. Lines 198-199: the authors stated that “We found that *Pto* MucD was localized to the membrane and was secreted outside of the cell (Figure S5B)”. The statement seems contradictory. Localized to the membrane or secreted outside of the cell?

(Response 11)

Thank you for pointing this out. Now we have modified the text to deliver our message more accurately. Now it reads

“We found that *Pto* MucD localized to the membrane but was also secreted outside of the cell (Supplementary Fig. 5A) as in *P. aeruginosa* (Damron et al., 2011; Okuda et al., 2011).”

We have also modified a similar description in Discussion. Now it reads

“Likewise, *Pto* MucD localizes to the membrane but is also secreted outside of the cell.”

8. Different sizes of MucD and its cleavage products were shown in the different figures. In Fig 3E, MucD-His was much smaller than 50 kDa and the cleaved MucD-His was about 30 kDa. However, in the other figures, MucD was about 50 kDa and the cleavage product was 37 kDa (such as in Fig 4B).

(Response 12)

We apologize our mistakes and thank the reviewer for pointing this out. When we prepared immunoblotting figures, indications of markers were misaligned. Now we have corrected these misalignments in relevant figures.

Reviewer #3 (Remarks to the Author):

This manuscript reports a well-performed series of experiments that convincingly show that site-specific cleavage of MucD by two homologous secreted aspartic proteases increases immunity against bacteria. I like this manuscript because it includes a large amount of carefully performed experiments that clearly demonstrate the direct link between a host protease and pathogen substrate. Especially the use of a non-cleavable substrate mutant, showing that this enhances virulence irrespective of the protease is neat. This story is well-suited for this journal.

I do, however, have two minor main points and several minor points:

1) Fig1A shows OD, but it is unclear if the higher OD is caused by stuff accumulating in the apoplast of flg22-treated plants or caused bacterial growth. Please show the number of live bacteria, as used in the rest of the manuscript, or show a t=0.

(Response 13)

The addition of apoplast fluid from flg22-treated leaves into bacterial culture led to lower OD as compared to control, likely due to reduced bacterial growth. Thus, even though flg22-treated apoplast fluid had contained chemicals, which might increase OD, decreasing bacterial OD with flg22-treated apoplast should not have been caused by this.

2) I am puzzled by the quick selection of two putative cleavage sites for an uncharacterized protease. Please explain more carefully how these sites were selected.

(Response 14)

We are also intrigued by the finding that both putative cleavage sites in MucD are under positive selection within *Pseudomonas*. As SAP homologs are highly conserved in angiosperms as we discussed in Discussion and microbes likely co-evolved with plants, this selection pressure might be higher than we think. While MucD F106 is likely targeted by *Arabidopsis* Col SAP1 and SAP2, MucD S394 might be targeted by SAPs from other plant species or by other plant aspartic proteases. We hope to address this important question in the future.

Minor points:

1) I think it should be mentioned that MucD is a virulence factor in both title and abstract. The way it is written now is neutral.

(Response 15)

mucD was required for *in planta* growth, suggesting that it is a virulence factor but was also required for general growth *in vitro* (Figure 4D). Therefore, we did not define MucD as a virulence factor in Title and Abstract.

2) L107 write ‘Immunoblotting of the SAP1-GFP fusion...’

(Response 15)

Thank you for suggestion. We have changed it to

“Immunoblotting of the SAP fusion proteins showed slightly increased apoplastic accumulation upon *Pto* infection (Fig. 1c).”

3) FigS2D should be a main figure.

(Response 16)

Thank you for suggestion. We have moved a part of Figure S2D to new Figure 3D. Accordingly, following panels in Figure 3 have been shifted.

4) Explain better why MucD was selected from 21 SAPTs

(Response 17)

Thank you for suggestion. We have modified the text. Now it reads

“The top SAPT candidate containing a putative aspartic protease digestion site(s) was MucD, which is an HtrA-like protease involved in the regulation of alginate biosynthesis and in the responses to heat and oxidative stress (Hay et al., 2012; Yorgey et al., 2001).”

5) L181 ‘SAP1 and SAP2 are required to cleave’

(Response 18)

We suspect that the reviewer pointed out for L191 instead of L181. We have modified the text. Now it reads

“These results indicate that SAP1 and SAP2 are required to cleave *Pto* MucD during infection.”

6) L308 states that ‘we did not address where SAP1 and SAP2 cleave MucD’. What do you mean? Is it not better to state that you have not ‘confirmed that cleavage occurs in the apoplast’?

(Response 19)

Thank you for suggestion. We have deleted the following sentences.

~~“Yet, questions remain. For instance, we did not address where SAP1 and SAP2 cleave MucD.”~~

7) M&M L370 states that apoplastic extraction buffer contains a protease inhibitor cocktail. Would that not interfere with protease assays done throughout the manuscript?

(Response 20)

We apologize the confusion. We used the protease inhibitor cocktail for apoplast protein extraction, but we did not use the protease inhibitor cocktail when we extracted apoplast fluids that were used in Figure 1A. This is the only experiment in which apoplast fluid was used. We have added a new section in Method which describes how we extracted apoplastic fluid.

8) Would it be possible to add stats to Fig1D (confocal images?)

(Response 21)

Figure 1D showed qualitative information. For instance, RFP signals were detected in between GFP signals (plasma membrane marker). Thus, we do not think that statistical analysis is necessary.

9) Why was Fig2H not part of Fig2B, like Fig2E and 2F?

(Response 22)

We placed Figure 2H in a different panel from Figure 2B because Figure 2B and 2H came from separated experiments.

10) Why was Fig3A not combined with Fig3C? SAP1 seems identical between the two.

(Response 23)

Figure 3A and Figure 3C came from separated experiments. In fact, the protease activities of SAP1 in these experiments were slightly different.

11) Why were the bacterial growth assays done with boiled SAP1 and not with the SAP1 mutant or pepstatin?

(Response 24)

We performed *in vitro* bacterial growth assay with a SAP1 mutant (Previous Figure S2D; now we have moved a part of data as Figure 3D). Pepstatin A is a general inhibitor of aspartic proteases. Thus, we did not perform experiments using pepstatin A as it can inhibit not only SAP1 but also other bacterial aspartic proteases, which does not allow us to conclude that SAP1 inhibits bacterial growth.

12) The readability of the figures would improve if it stated *in vitro*/*in vivo* or some other info.

(Response 25)

Thank you for suggestion. We have included such information in figures.

13) Fig5C is printed too small to read.

(Response 26)

Thank you for suggestion. We have enlarged Fig5C.

Reviewers' comments:

Reviewer #1 (Remarks to the Author):

I have read the responses of the corresponding author to my comments and those made by the two other reviewers. I also read recent articles on the function of the MucD serine protease in *Pseudomonas aeruginosa*. I don't think MucD can be called a virulence factor for Pto as the MucD mutant gives a complete different phenotype in vitro when compared to wild type (see SFig. 5A). The exopolysaccharide composition/production of the mutant of MucD and the inactive MucD (active site mutation) likely differ significantly. However, breakdown of SAP1/SAP2 by MucD is positively correlated with increased bacterial growth in planta, although bacterial growth was followed only for a very short time. My questions on infections assays and statistics are adequately answered

Reviewer #2 (Remarks to the Author):

Additional experiments to address some of my concerns would be nice, but I understand that it would take a quite long time to carry out some of these experiments. Most of my concerns have been addressed seasonally well, so I consider that the revision is adequate.

Reviewer #3 (Remarks to the Author):

I thank the authors for so quickly returning their comments on my queries. Most of them have been addressed adequately but my two main questions were apparently misunderstood. So I ask them again in a different way.

- 1) The authors claim that bacteria grow less in AF from flg22-treated plants based on OD measurement (Fig1A). This is not an adequate measurement to measure bacterial growth. This should be done by plating out the bacteria and determine colony-forming units.
- 2) How is it possible that the specificity for two cleavage sites in MucD is predicted for a non-characterized protease? Please describe this more carefully in L200.

Point-to-point response letter

Reviewer #1 (Remarks to the Author):

I have read the responses of the corresponding author to my comments and those made by the two other reviewers. I also read recent articles on the function of the MucD serine protease in *Pseudomonas aeruginosa*. I don't think MucD can be called a virulence factor for *Pto* as the MucD mutant gives a complete different phenotype in vitro when compared to wild type (see SFig. 5A). The exopolysaccharide composition/production of the mutant of MucD and the inactive MucD (active site mutation) likely differ significantly. However, breakdown of SAP1/SAP2 by MucD is positively correlated with increased bacterial growth in planta, although bacterial growth was followed only for a very short time. My questions on infections assays and statistics are adequately answered.

(Response 1)

Thank you for the thoughtful comment. We agree to the comment and we think that we already avoided defining *Pto* MucD as a virulence factor in the previous version of manuscript.

Reviewer #2 (Remarks to the Author):

Additional experiments to address some of my concerns would be nice, but I understand that it would take a quite long time to carry out some of these experiments. Most of my concerns have been addressed seasonally well, so I consider that the revision is adequate.

(Response 2)

Thank you for helping us improve the manuscript.

Reviewer #3 (Remarks to the Author):

I thank the authors for so quickly returning their comments on my queries. Most of them have been addressed adequately but my two main questions were apparently misunderstood. So I ask them again in a different way.

1) The authors claim that bacteria grow less in AF from flg22-treated plants based on OD measurement (Fig1A). This is not an adequate measurement to measure bacterial growth. This should be done by plating out the bacteria and determine colony-forming units.

(Response 3)

As requested by the editor and the reviewer, we have performed a new experiment for Fig. 1a by counting cfu instead of measuring OD. We obtained a very similar result with the previous one. We have replaced the previous Fig. 1a with a new Fig. 1a.

2) How is it possible that the specificity for two cleavage sites in MucD is predicted for a non-characterized protease? Please describe this more carefully in L200.

(Response 4)

We have changed the description. Now it reads

“We generated MucD^{F106Y} and MucD^{S394A} with a mutation at each of the putative cleavage sites by aspartic proteases and produced His-tag-fused recombinant proteins at the C-terminus. In the *in vitro* cleavage assay, we observed that SAPI-GST cleaves MucD^{S394A} but not MucD^{F106Y} (Fig. 4b).”